**Subject Category:**
Biology (whole organism)

acoustics/biomechanics/physiology

crocodilians, head-related transfer function, bioacoustics, sound localization

**Author for correspondence:**
L. Papet
e-mail: leo.papet@inserm.fr

# Influence of head morphology and natural postures on sound localization cues in crocodilians

## L. Papet[1,2], N. Grimault[1], N. Boyer[2] and N. Mathevon[2]

[1]Centre de Recherche en Neurosciences de Lyon – Equipe Cognition Auditive et Psychoacoustique, CNRS UMR 5292, Univ. Lyon 1, Lyon, France
[2]Equipe Neuro-Ethologie Sensorielle ENES/NeuroPSI, CNRS UMR 9197, University of Lyon, Saint-Etienne, France

LP, 0000-0001-8916-4666; NM, 0000-0003-0219-6601

As top predators, crocodilians have an acute sense of hearing that is useful for their social life and for probing their environment in hunting situations. Although previous studies suggest that crocodilians are able to localize the position of a sound source, how they do this remains largely unknown. In this study, we measured the potential monaural sound localization cues (head-related transfer functions; HRTFs) on alive animals and skulls in two situations, both mimicking natural positions: basking on the land and cruising at the interface between air and water. Binaural cues were also estimated by measuring the interaural level differences (ILDs) and the interaural time differences (ITDs). In both conditions, HRTF measurements show large spectral variations (greater than 10 dB) for high frequencies, depending on the azimuthal angle. These localization cues are influenced by head size and by the internal coupling of the ears. ITDs give reliable information regarding sound-source position for low frequencies, while ILDs are more suitable for frequencies higher than 1.5 kHz. Our results support the hypothesis that crocodilian head morphology is adapted to acquire reliable localization cues from sound sources when outside the water, but also when only a small part of their head is above the air–water interface.

## 1. Introduction

As top predators, crocodilians have developed fascinating sensory skills: accurate vision in air [1,2], highly developed olfaction [3,4], precise abilities to detect water vibrations [5–7] and an acute sense of hearing [8,9]. Experimental studies, as well as field observations, have demonstrated that the auditory modality is

of primary importance for both their social life (e.g. during mating and mother–offspring interactions) and hunting success [10,11]. Although it is known that crocodilians hear sounds over a broad frequency range (from around 300 Hz up to 8 kHz, with a peak in audiogram around 1 kHz [8,9]), and while the functional anatomy of the ear of these archosaurs is similar in many respects to that of birds [12], we have little knowledge on many aspects of their hearing biology. Thus, while some observations have brought evidence for directional hearing in crocodilians, suggesting the existence of specialized adaptations [13–15], how they localize sound sources in their environment remains poorly described.

In birds and mammals including humans, sound-source localization relies on interaural time differences (ITDs; a sound from the left will arrive at the left ear first) and interaural level differences (ILDs; a sound from the left will be louder in the left ear because of the shadowing effect of the head). Moreover, monaural spectral cues are induced by the filtering effect of the head: head-related transfer functions (HRTFs) result from sounds experiencing spectral modifications during their propagation through and around the head, providing different spectral cues when arriving from different azimuths and elevations [16]. The detection of sounds and the perception of localization cues are often reinforced by some anatomical particularities such as the external ears of mammals [17] and ruff feathers and asymmetrical ears of the barn owl [18]. Remarkably, the hearing apparatus of crocodilians includes a well-developed external ear formed by horny, prominent bone overhanging a muscular ear-lid that protects the eardrum [15,19,20], suggesting a functional role in directional hearing.

Previous studies showed that some directional information is encoded in the auditory nerve of juvenile American alligators *Alligator mississippiensis*. These data support the hypothesis that the acoustic coupling of middle-ear air cavities—an anatomical particularity found in both birds and crocodilians—could enhance localization abilities [21]. By combining results from passive acoustic experiments and measurements of auditory brainstem response to sounds in young alligators, Bierman *et al.* [21] suggested that HRTFs do not entirely account for the level of directional sensitivity in auditory nerve activity. However, this requires further consideration and the current study will focus on larger animals (alive and skulls), in natural positions in water, and on an extended frequency range.

The main characteristic of crocodilian biology is their amphibious way of life. These animals spend most of their active time at the interface between air and water while they mostly come on land to regulate their internal temperature by basking motionless in the sun [22]. In water, the upper part of the head—with the nostrils, eyes and ears—appears above the waterline (figure 1*a,b*). The acoustic environment of crocodilians may be strongly influenced by this position, as air and water show different sound-propagation properties. As water acts as a reflective surface, it should contribute to the properties of sound waves arriving at the eardrums.

In the present paper, we investigate the acoustic localization cues available to crocodilians, paying specific attention to the effect of the air–water interface and its interaction with head morphology and head size. We hypothesize that localization cues generated by interactions between sound waves and the head morphology still exist when the animal is at the interface of air and water, in spite of having most of its body concealed underwater. It is worth noting that the presence of such cues does not necessarily induce the behavioural use of the cues by the animals. Using microphones positioned within the ears, we recorded sounds emitted by a source set up at different azimuths from the animal's head axis. We then measured HRTFs, which characterize the transfer function of the spatial acoustic filter created by the head and body of the animal. This method has commonly been used in humans and other species (e.g. mammals [23,24], alligators [21] and birds [18]). As HRTFs are related to the complex absorption and reflection patterns of acoustical waveforms by the head and body, we compared HRTFs obtained with the crocodile positioned at the interface of air and water with those obtained with the animal's whole body outside water. We further estimated the potential influence of species-specific head shape on HRTFs by performing the experiment on individuals from two different species: the Nile crocodile *Crocodylus niloticus* and the broad-snouted caiman *Caiman latirostris*. To assess the impact of head size, we measured HRTFs on three skulls of different rostral snout lengths: 6.9 cm, 16.5 cm and 22.7 cm, corresponding, respectively, to a juvenile, a young and a subadult Nile crocodile. Beside HRTF monaural cues, the binaural cues ITDs and ILDs are also well known to facilitate sound-source localization. We computed ITDs by comparing the waves' arrival time between the two ears, and ILDs by calculating the difference in sound pressure level measured at the right and left ears. Overall, our results may suggest that the shape of the crocodilian head induces both monaural (HRTF) and binaural (ITD and ILD) sound localization cues even when the head is mostly concealed in water.

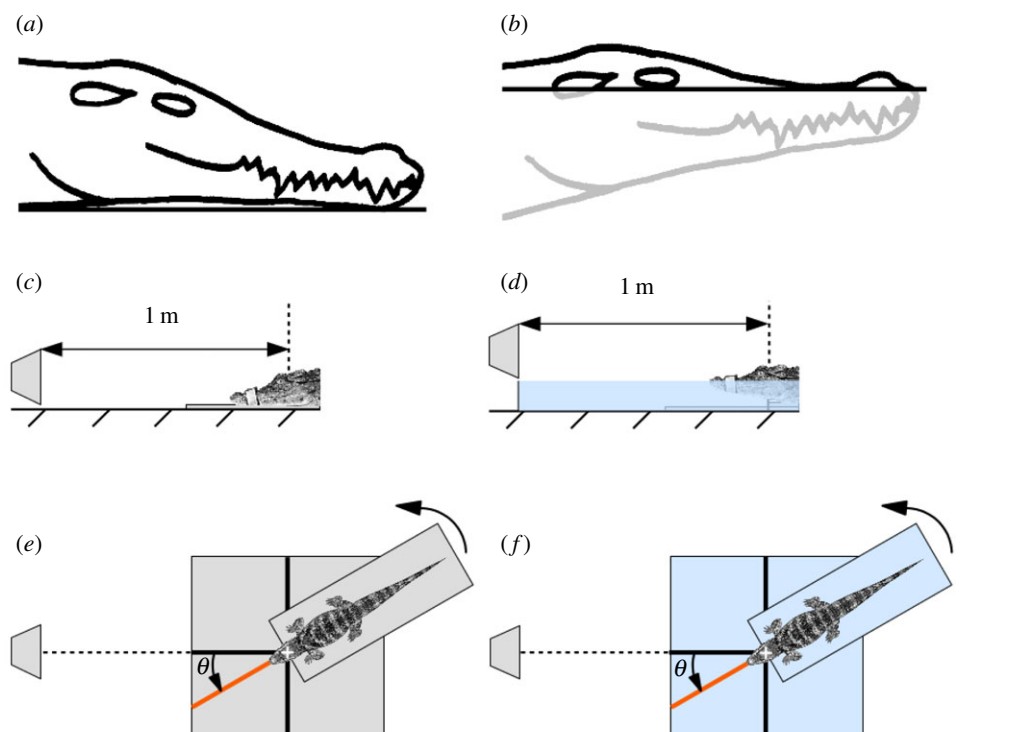

**Figure 1.** Experimental set-up used to measure HRTF localization cues. Two natural postures of crocodilians are considered in the present study: (*a*) on the land and (*b*) at the interface between air and water. Cross-section and top view of the set-up in the land condition (*c,e*), and at the interface (*d,f*).

# 2. Material and methods

## 2.1. Experimental models

We assessed acoustic localization cues on two living animals (figure 2) and three skulls (figure 3). The animals were one broad-snouted caiman *Caiman latirostris* (rostral snout length: 4.4 cm, 2 years old) and one Nile crocodile *Crocodylus niloticus* (rostral snout length: 6.9 cm, 2 years old), provided by the zoo Planète Crocodiles (Civaux, France). Both individuals were accommodated in the ENES laboratory in Saint-Étienne, France, in dedicated areas. These species show strong differences in head morphology [25] that are well illustrated by the rostral proportion: *C. latirostris* has the broadest snout, whereas the rostral proportion of *C. niloticus* is just above the average of the 23 crocodilian species. In addition, we measured monaural and binaural cues on three skulls of Nile crocodiles (rostral snout lengths of 6.9 cm, 16.5 cm and 22.7 cm). More biometric data are detailed in the electronic supplementary material.

## 2.2. Animal condition

A critical point with the living individuals was to prevent movements while minimizing stress during HRTF measurements. Three months prior to the experiment, both animals were habituated to remain motionless on a board for 30 min. The animal position was further secured by straps. This procedure allowed anaesthesia to be avoided, as it is difficult to administer in crocodilians and could have been a survival risk [22]. During the experiments, we continuously assessed the stress level of the animal by observing its pupillary dilatation and behaviour (escape attempts). During the weeks following the experiments, the condition of both animals was carefully monitored in terms of growth parameters and behaviour (e.g. food intake), and the animals behaved similar to the time before testing.

## 2.3. Signal acquisition for HRTF measurements

We measured HRTFs under two conditions mimicking biologically relevant situations (figure 1), as follows.

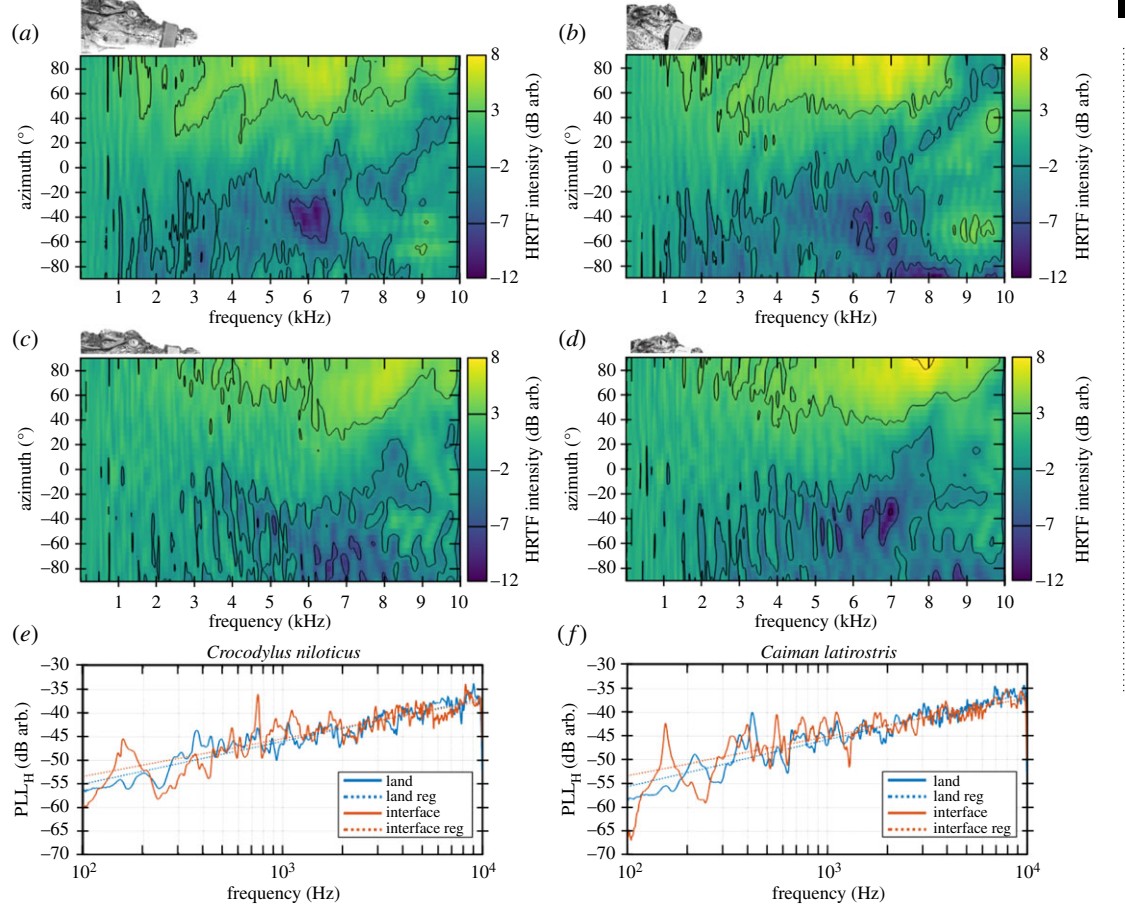

**Figure 2.** HRTF measured on awake animals in two natural positions. (*a*) HRTF measured on *Crocodylus niloticus* in the land condition. (*b*) HRTF measured on *Caiman latirostris* in the land condition. (*c*) HRTF measured on *Crocodylus niloticus* in the interface condition. (*d*) HRTF measured on *Caiman latirostris* in the interface condition. The considered animal and configuration are represented above each panel at a scale of 1/12. (*e*) Potential localization level (PLL) measured on *Crocodylus niloticus* in land (blue) and interface (red) conditions. (*f*) PLL measured on *Caiman latirostris* in land (blue) and interface (red) conditions. (*e,f*) Solid lines correspond to raw data and dashed lines are linear regressions.

(1) 'Land condition': the animal was placed on a board lying on the land of a semi-anechoic room (LVA-INSA Lyon: background noise level = $20 \pm 1$ dB SPL; reverberation time = $0.1 \pm 0.1$ s), mimicking a position frequently used by crocodilians when basking (figure 1*a,c,e*). The ground of the semi-anechoic room can be considered as perfectly reflective to acoustic waves.

(2) 'Interface condition': the animal was placed in water, with its nostrils, eyes and ears just above the waterline (figure 1*b,d,f*). This condition mimicked the natural position of an animal in water, e.g. when cruising, ambushing prey or protecting its young. In this position, the water surface was also considered to be fully reverberant, owing to the short distance between the acoustic source and the microphones.

Under both conditions, the sound source (loudspeaker) was positioned 1 m from the centre of the head of the animal (defined as the point equidistant between the two ears; see figure 1), with $0°$ elevation. A rotation of the animal's body along its antero-posterior axis enabled measurements of HRTFs in a two-dimensional plane between $-90°$ and $+90°$ in $5°$ increments. The sound-emitting equipment was composed of a sound card (Presonus, Audiobox 44-VSL) connected via an amplifier (Yamaha, AX-397) to the loudspeaker (AudioPro loudspeaker, Bravo Allroom Sat). The loudspeaker was hung just above the ground or the waterline to avoid acoustic coupling with the ground or the water. The centre of the medium loudspeaker membrane was placed at the same height as the microphones that were inside the ears.

The experiments were always performed with the ears out of the water. We placed one microphone (Knowles, FG-23329-P07) inside the cavity of each ear, behind the ear-lid and close to, but without

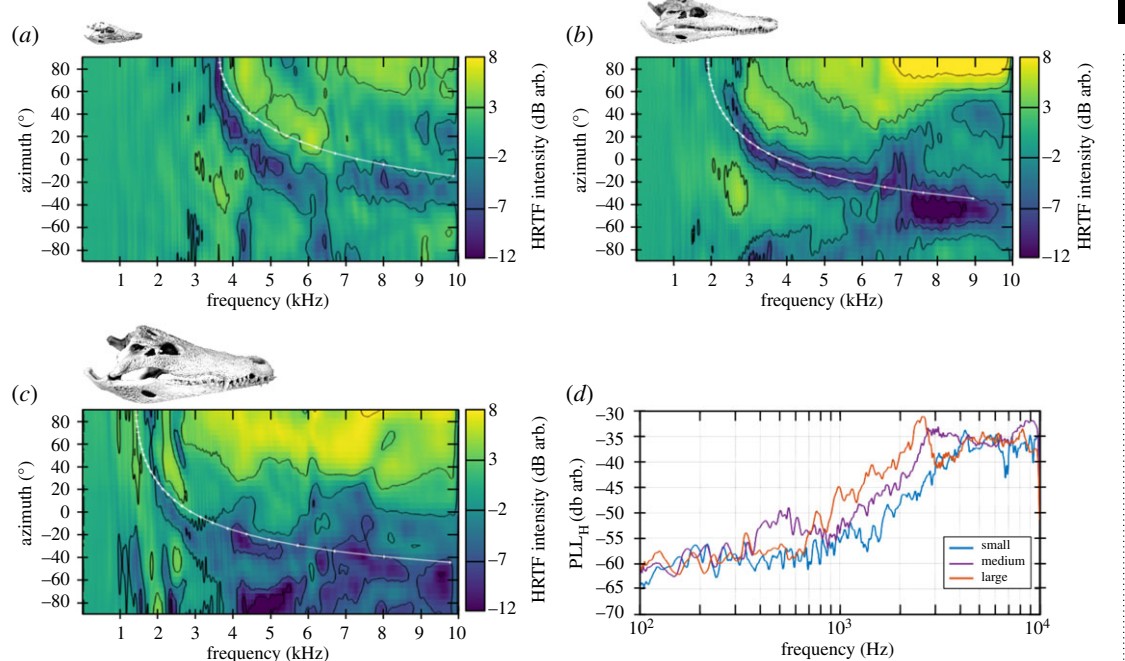

**Figure 3.** HRTFs measured on three skulls of different sizes. ($a-c$) HRTFs measured on three *Crocodylus niloticus* skulls of different lengths: 6.9 cm, 16.5 cm and 22.7 cm, respectively. The solid white line represents the destructive interferences based on a simple geometrical model of the path difference. The considered skull is represented above each panel at a scale of 1/12. ($d$) PLL computed for the three skulls: 6.9 cm (blue), 16.5 cm (purple) and 22.7 cm (red).

touching, the tympanic membrane. This allowed simultaneous recording of the sounds arriving at both right and left ears. The recording equipment was composed of both left and right microphones connected to the input channels of the sound card (sampling frequency = 44.1 kHz).

The emitted signal was a logarithmic sine sweep (frequency range = 20–10 000 Hz; duration = 5 s; intensity level = 80 ± 0.5 dB SPL). The frequency range was chosen to mostly cover the hearing range of crocodilians, which is mainly centred on 1–2 kHz [8,9]. For analysis purposes, we used a Matlab code to synchronize in time the source signal with the two recorded microphone signals.

Prior to each measurement, we calibrated the broadcast signals in the absence of any animal, with both microphones placed at the virtual centre of the crocodile's head. This calibration was a necessary step to take into account the properties of the sound-producing and recording equipment ('transfer function' due to material gains, frequency responses, etc.) in HRTF measurements. Recorded signals were averaged ($n = 10$) and used as a reference to compensate for this transfer function.

In addition to HRTF calculations, we measured ITDs and ILDs using the out-of-water set-up. First, we sent an impulse signal (pulse) to estimate the broadband ITD. We then broadcast 500 ms pure tones at 125, 250, 500 and 1000 Hz. Since Carr *et al.* [26] demonstrated that in alligators ITDs are not coded in the auditory nerve for frequencies over 1 kHz, we chose to consider only frequencies below 1 kHz. The ITD corresponds to the value of $\tau$ maximizing the cross correlation between left and right ears $\int_{-\infty}^{+\infty} s_L(t)s_R^*(t - \tau)\,dt$, where $t$ is the time and $\tau$ is the time delay between left and right microphone signals ($s_L(t)$ and $s_R(t)$, respectively). Assuming symmetry of the head, ITDs were normalized to the $0°$ value. Directly from HRTF measurements and because of the assumption of symmetry to the normal incidence, the ILD can be calculated for a given frequency and azimuth as

$$\mathrm{ILD}(f, \theta) = H(f, \theta) - H(f, -\theta), \tag{2.1}$$

where $f$ is the frequency, $\theta$ is the azimuth of the sound source and $H(f, \theta)$ is the HRTF.

## 2.4. Signal processing

To avoid clipping, we applied a Hanning ramp (501 points) at the onset and at the end of recorded microphone signals, and normalized all recorded signals by the root-mean square amplitude of the normal incidence signal (left and right channels independently).

The spectrum of the recorded microphone signals ($R$) within the ear can be expressed as a linear combination of frequency ($f$), sound-source azimuth ($\theta$), elevation ($\phi$) and microphone position ($\vec{x}$), such as

$$R(f, \theta, \phi, \vec{x}) = S(f) \times H(f, \theta, \phi) \times \mu(f, \vec{x}), \tag{2.2}$$

where $S(f)$ is the calibration signal, $H(f)$ is the HRTF and $\mu(f, \vec{x})$ is the contribution of the microphone position (adapted from [16]).

In our experiments, elevation was maintained at $0°$. HRTFs thus depend only on the sound-source azimuth and the sound frequency. One caveat concerns the in-ear position of the microphones: as they were placed under the (opaque) ear-lids, their position could not be perfectly assessed and could be slightly different between left and right ears. We took into account this potential issue by performing two methodological steps. First, the position of the microphone was carefully controlled to be as reliable as possible. Second, we used a normalization method adapted from that developed by Middlebrooks & Green [16] to reduce the effect of the microphone position in the human ear canal when measuring HRTF. Briefly speaking, the measured microphone signal was normalized by the average of all microphone signals ($\mu(f, \vec{x})$), depending only on the frequency and the position of the microphone. The HRTF was then supposed to depend only on the frequency and the azimuth of the sound source, as follows:

$$H(f, \theta) = \frac{R(f, \theta, \vec{x})}{S(f) \times \mu(f, \vec{x})}, \quad \text{with: } \mu(f, \vec{x}) = \frac{1}{S(f) \times N} \sum_{i=1}^{N} R_i(f, \theta_i, \vec{x}), \tag{2.3}$$

where $N$ is the number of microphone signals.

To limit the error in HRTF estimations, we considered in this study each head as symmetrical and thus averaged the HRTFs simultaneously measured in the right and left ears. To limit discontinuities in HRTF measurements along the angular and frequential axes, we applied a smoothing procedure based on a two-dimensional floating Gaussian window normalized in amplitude with a five point width in azimuth and a logarithmically varying width in frequency (3 Hz at $f = 20$ Hz and 1 kHz at $f = 10$ kHz).

Finally, we calculated a potential localization level (PLL) based on HRTF and ILD as follows:

$$\text{PLL}_{\text{H}}(f) = 20 \times \log_{10}\left(\frac{1}{N_\theta} \sum_{\theta=-90}^{90} \left|\frac{\partial H(f, \theta)}{\partial \theta}\right|\right) \tag{2.4}$$

and

$$\text{PLL}_{\text{ILD}}(f) = 20 \times \log_{10}\left(\frac{1}{N_\theta} \sum_{\theta=-90}^{90} \left|\frac{\partial ILD(f, \theta)}{\partial \theta}\right|\right), \tag{2.5}$$

with $N_\theta$ is the number of azimuth positions (here $\theta$ varies between $-90°$ and $+90°$ with a step of $5°$, so: $N_\theta = 37$). $\text{PLL}_{\text{H}}(f)$ and $\text{PLL}_{\text{ILD}}(f)$ are expressed in dB. The PLLs are computed to facilitate the comparison of HRTFs and ILDs between the different conditions. Classically, sound localization cues are considered efficient when varying monotonously according to the azimuth of the sound source. For each frequency, the PLL corresponds to the average of the variation of $H(f, \theta)$ or $ILD(f, \theta)$ according to $\theta$ in dB. So, the PLL is a cumulated measurement of variations of monaural cues across the azimuth and a high PLL indicates a strong variation of the considered cue with the position of the source.

# 3. Results

## 3.1. HRTF cues in land and air−water interface conditions

The HRTFs measured in the awake animals in both land and interface conditions are displayed in figure 2a−d. The HRTF intensity level is coded by an arbitrary coloured dB scale (from $-12$ to $+8$ dB arb., with contour lines representing 5 dB intervals), and expressed as a function of both frequency (20−10 000 Hz) and sound-source azimuth ($-90°$ to $+90°$). Positive (respectively, negative) values of HRTF intensity level induce an amplification (respectively, attenuation) of the acoustic field owing to the presence of the head of the animal compared with the calibration situation (i.e. with no animal). Positive azimuth angles correspond to sounds recorded from the ipsilateral side, i.e. from the side of the sound source, whereas negative azimuths correspond to the contralateral side, i.e. sounds recorded in the 'acoustic shadow' of the head.

In both land and air–water interface conditions, HRTFs showed similar complex patterns of sound pressure level variations, with high dynamics (20 dB) depending on both sound-source incidence and sound frequency (figure 2a–d). This intensity pattern varied depending on the side: for azimuths smaller than 0° (contralateral side), the sound level measured near the eardrum is negative whereas it appears mainly positive (i.e. amplified) for incidences larger than 0° (ipsilateral side). This main result indicates that the angular position of the sound source influences the spectral cues perceived at the level of the ears, suggesting that HRTFs could support sound-source localization in both the land and air–water interface conditions. The difference in sound-pressure levels between the ears due to the position of the sound source was, however, mainly present for frequencies above 1 kHz. Below 400 Hz, the sound-pressure level was mostly invariant to sound-source position.

Beside this general picture, HRTFs were characterized by 'bumps' and 'notches', which may increase the locatability of the sound source. Globally, the complexity of the HRTF patterns increases with frequencies (figure 2). As the emitted sound signal showed equal levels across the whole frequency spectrum, the variations of this level are due to the filtering effect of the head. These complex shapes may underlie the complexity of the acoustic field surrounding the animals' head. For instance, when the Nile crocodile was in the land condition (figure 2a), the sound-pressure level near 6 kHz was strongly influenced by the source angle (variations from −11 dB up to +6 dB), with a marked area of low levels when the source was positioned at −40°.

Overall, these results suggest that HRTF localization cues are already present near 1 kHz in both land and interface conditions, and become more important when sound frequency increases. This is further illustrated by the PLLs displayed in figure 2e,f. PLLs represent the amount of external acoustic localization cues measured at the tympanic membrane level (see Material and methods). In accordance with HRTF results, PLLs increase exponentially with sound frequency (i.e. linearly with the logarithm of the frequency). They look very similar in both species and conditions (land condition: 2.7 dB/octave for the crocodile and 2.9 dB/octave for the caiman; air–water interface condition: 2.4 and 2.5 dB/octave, respectively).

## 3.2. Impact of skull size and acoustic coupling through the interaural canal

The HRTFs measured on the three skulls of *Crocodylus niloticus* are displayed in figure 3a–c. Compared with those obtained in live animals, HRTFs show mainly the same pattern with more complex cues in high frequencies and a higher intensity level in the ipsilateral side. In the low-frequency range ($f <$ 1 kHz), it can be seen that HRTF patterns appear more homogeneous in contrast to those measured on live animals. Nevertheless, the main difference is the presence of a 'crescent-shaped' area of low sound-pressure level (highlighted by the continuous white line on the HRTF colour maps in figure 3a–c). This pattern, consistently found in the three skulls, is included in a frequency band which is directly dependent on the skull size (3.5–6.5 kHz, 2.5–6.5 kHz and 1.7–5.5 kHz for the small, medium and large skulls, respectively), and may be due to destructive interferences caused by the interaural canal. Considering one ear, the difference in the pathway between the direct wave and the wave arriving from the other ear through the interaural canal induces a phase difference $\Phi$,

$$\Phi = 2\pi f \frac{\delta}{c}, \tag{3.1}$$

where $\delta$ is the difference in the pathway in metres, $c$ is the sound velocity and $f$ is the frequency. From (3.1), if $\Phi = \pi[2\pi]$, we can compute the frequencies corresponding to destructive interferences inside the interaural canal $f_{ic}$,

$$f_{ic} = \frac{c}{2\delta}. \tag{3.2}$$

This simple geometrical model plotted in figure 3a–c indeed suggests that this pattern is due to the interaural canal.

In line with what we found in live animals, the effect of sound-source position on HRTFs varied as a function of skull size. For instance, a 2 kHz sound produces complex level variation that depended on the sound-source position in the larger skull (between −4 and +4 dB), while the sound-pressure level remained constant for all azimuths (0 dB) for the small and medium skulls. Moreover, the maximum sound-pressure level areas measured on the ipsilateral side extended to a lower frequency range when skull size increased. For an incidence of 90°, the 3 dB contour line starts at 4040 Hz, 3050 Hz and 2120 Hz for the small, medium and large skulls, respectively.

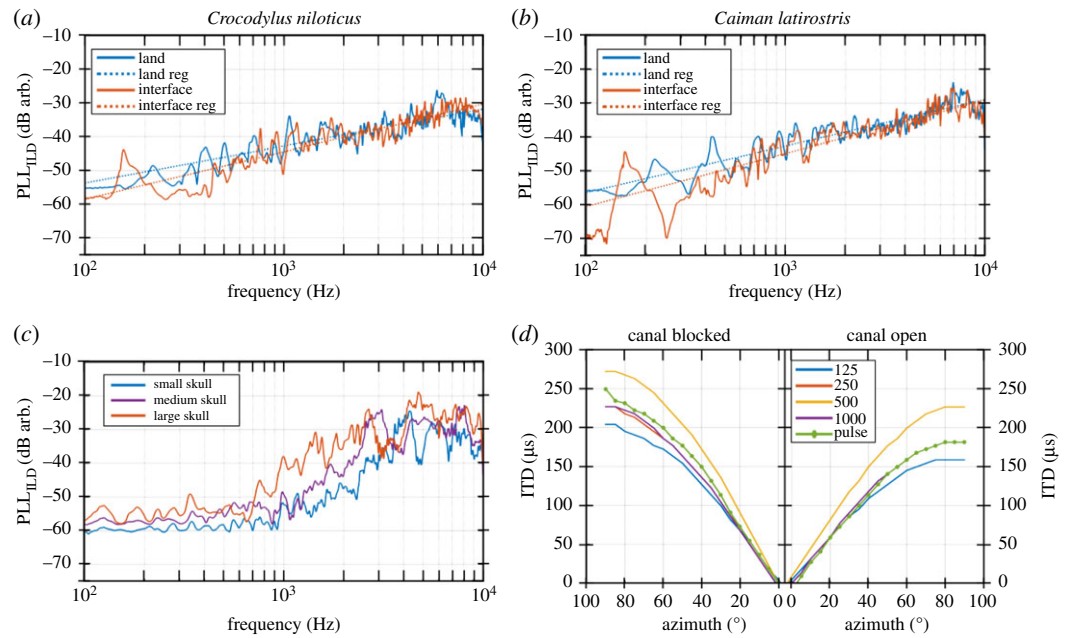

**Figure 4.** Binaural cues measured on awake animals and skulls of different sizes. PLL computed on ILD for *Crocodylus niloticus* (a) and *Caiman latirostris* (b) measured in the land (blue) and interface (red) situations. Dashed lines in (a) and (b) correspond to linear regressions. (c) PLL computed for ILD for three different-sized Nile crocodile skulls: 6.9 cm (blue), 16.5 cm (purple) and 22.7 cm (red). (d) ITD measured for four pure tones (125, 250, 500 and 1000 Hz) and for a pulse with the interaural canal blocked (left) and open (right).

In skull measurements, the PLLs did not increase linearly with the logarithm of the frequency (figure 3d), and it is not relevant to model its evolution using linear regression. In the low-frequency range ($f <$ 1 kHz), the PLLs remained almost steady around $-60$ dB. For frequencies larger than 1 kHz, the PLLs increased with frequency in line with the complexity of HRTF patterns. In skulls, HRTFs depended on the global shape of the head but were also modified by the interaural canal, causing a nonlinear evolution of the PLLs.

## 3.3. Binaural cues

Based on equation (2.1), PLLs were computed from ILDs and calculated using (2.5) (figure 4a,b). As displayed in figure 4a–c, the PLLs calculated for live animals increased monotonically with the logarithm of frequency, with no noticeable impact of species or condition. The effect of head size is emphasized by PLLs calculated from skulls. Thus for a 1 kHz sound, PLL is equal to $-52$, $-49$ and $-38$ dB for the small, medium and large skulls, respectively (figure 4c). ILDs are stronger for frequencies higher than 1 kHz, with a sudden increase in the slope near 1 kHz of PLLs measured on skulls.

We assessed ITDs on the medium-sized skull in only two conditions: the interaural canal was either obstructed with an adhesive or open (figure 4d). The ITD results are very close to those obtained by Carr *et al.* on *Alligator mississippiensis* [26]: ITDs are symmetrical to the normal incidence and vary monotonically with the position of the sound source. When open (figure 4d, right), the interaural canal led to a decrease in ITDs, offering a shorter pathway to acoustic waves. As a result, the maximum of ITD (at 90°) is decreased by about 50 µs when the canal is open, independently of the frequency.

# 4. Discussion

Our study presents evidence that the morphology of the head of crocodilians induces monaural and binaural acoustic cues that are available to the animal and that are potentially useful to obtain information on the position of a sound source. These cues are still present when most of the animal's body is underwater, suggesting that the well-developed external ear formed by the horny and prominent bone is efficient at providing external localization cues both on the land and at the

interface. This could represent an evolutionary adaptation to the peculiar amphibious behaviour of crocodilians.

Spectral monaural cues (HRTF) are present mainly for frequencies higher than 1 kHz. Their saliency increases with sound frequency, and they are strongly influenced by head size, with a shift to a lower frequency range in larger heads. Interestingly, we found that HRTF cues are very similar in both 'land' and 'air–water interface' conditions. This suggests that the ability to use monaural cues for sound-source localization may be alike under both conditions, despite only part of the crocodile's head being exposed in the interface condition. Our investigations on skulls underline the importance of the interaural canal, which creates destructive interferences that may further facilitate sound-source localization. In addition to highlighting HRTF cues, we confirm the presence of ILD cues (mainly for frequencies above 1 kHz) as well as ITD cues (mainly in the low-frequency range, below 1 kHz). Altogether, monaural and binaural cues may allow crocodilians to accurately localize the position of a broadband sound source in their environment.

Overall, our results are consistent with those of Bierman et al. [21]. These authors concluded from their measurements of external cues that the 'acoustic space cues generated by the external morphology of the animal are not sufficient to generate location cues that match physiological sensitivity' [21]. Indeed, they demonstrated that the level of physiological sensitivity is due to the contributions of the sound localization cues, the internal coupling of the middle ears and the directionality of the eardrum. However, Bierman et al. worked on very young alligators, with an interaural distance of $2.25 \pm 0.2$ cm, whereas our animals showed interaural distances of 3.9 cm and 4.7 cm, respectively, for the broad-snouted caiman and the Nile crocodile. The juvenile skull of Crocodylus niloticus had interaural distances comparable to those of the alligators studied by Bierman et al. (2.4 cm). In addition, they conducted their experiments for frequencies lower than 4 kHz and they obtained a maximum variation of 8 dB (for a frequency of 2 kHz). In the same frequency range, we measured a slightly higher dynamics of 9 dB maximum (cf. figures 2 and 3). In the present study, the frequency span is increased up to 10 kHz because, for high-intensity sounds, the high frequencies may be of interest for crocodilians. Indeed we cannot exclude the potential use of frequencies higher than 2 kHz, and, as we have strong dynamics (up to 20 dB), this frequency region may be relevant for sound localization. Assuming the small size of the animals and that the external cues are shifted to low frequency when size increases (figure 3), our results extend the work of Bierman et al. and is coherent with their results.

Given that the crocodilians' audiogram is centred on the lower part of the frequency spectrum (1–2 kHz [8,9,27]) and that the external localization cues increase with frequency, the biological relevance of the HRTFs has to be discussed. The frequency span used in our study was chosen to widely cover the auditory curve of crocodilians, and to illustrate the low-frequency shift phenomena induced by size. If we consider the maximum auditory sensitivity of crocodilians (i.e. 1.5 kHz), the dynamics measured on external cues is more than 5 dB (figure 2), which is potentially sufficient for sound localization. In blackbirds, pigeons and sea lions the minimum detectable binaural level difference is around 3 dB [28,29], whereas it is 1 dB in humans [30].

Our study focused on two awake juvenile animals belonging to two different species. However, the results could be extended to larger animals. Although we found a comparable amount of potential localization cues in both experimental animals, slight differences in HRTFs between species were visible in the upper part of the frequency spectrum. Studies in humans have shown that individual morphology (i.e. rather small morphological differences) may significantly influence HRTFs [31]. As both our animals had comparable body sizes, these differences may have been induced by their respective head morphology (e.g. the Nile crocodile has a much more slender snout than the caiman). In larger animals, these slight differences should be enhanced and shifted towards lower frequencies. As a consequence, they may significantly influence HRTFs within their hearing range.

Besides HRTFs, the internal acoustic coupling may increase sound localization because two waves arrive at the same eardrum: one from the outside and one from the inside through the interaural canal. The influences of this coupling were shown using a method of geometrical acoustics on skulls (figure 3). The interaural canal also influences the ITD by decreasing its value from 50 μs. While this result looks to be opposite to previously published data [32,33], it is explained by the fact that we consider the interaural canal on skulls but without any soft tissue (such as eardrums). The canal acts as a wave-guide without any obstacle to the propagation of the acoustic wave. Bierman et al. [21] demonstrated the implication of the directionality of the eardrums in the physiological computation of ITDs thanks to a laser vibrometry method.

ITDs are large in the low-frequency region (from 100 to 1000 Hz), whereas ILDs predominate in the high-frequency region ($f > 1500$ Hz). In his duplex theory developed for humans [34], Lord Rayleigh

asserted that ITDs and ILDs ensure strong localization abilities across the full auditory spectrum because of an overlap of both binaural cues between 1 and 1.5 kHz [35]. This theory cannot be applied to all vertebrates: as an example, the barn owl combines ITD and ILD information in the same frequency range (between 3 and 10 kHz) to localize, respectively, the azimuth and the elevation [36,37]. In another way, crocodilians might also qualify this theory. Carr *et al*. [26] demonstrated that ITDs are not neurally processed for frequencies higher than 1 kHz because of a lack of phase locking. As a consequence, ITDs and ILDs are weak in the band between 1 and 1.5 kHz, creating a lack of localization cues in this frequency region, at least for juvenile and medium-sized crocodilians. The maximum vibration of the eardrum was measured in this precise bandwidth [21], which could be used to compensate this 'gap' of localization.

As a conclusion, our results establish a strong background regarding the acoustic cues available to crocodilians when they localize a sound source in their environment—a frequent situation in several behavioural contexts, from predation to caring for the young. Our study focused on aerial hearing. However, crocodilians are amphibious animals and previous studies suggested a fairly good underwater hearing sensitivity. It would thus be interesting to investigate sound localization cues in an underwater context. Finally, sound localization abilities also remain poorly investigated and, even if a few behavioural observations in crocodilians have been published [13,38], the behavioural relevance of sound localization cues needs to be tested in subsequent research.

Ethics. All the experiments were conducted under the ENES lab agreement number (D 42-218-0901). The animals were under the responsibility of Nicolas Boyer, who is officially certified to work with crocodilians (certificate of capacity).
Data accessibility. The datasets and program files supporting this article have been uploaded on the Zenodo platform (zenodo.org/record/2572624#.XGv8UbhCdQI [39]).
Authors' contributions. L.P. designed the study, conducted the experiments and ran the analysis. N.G. and N.M. participated in data collection and analysis. N.B. participated in data collection and was in charge of the manipulation and of the well-being of the animals before, during and after the experiments. L.P., N.G. and N.M. wrote the manuscript. All authors gave final approval for publication and agree to be held accountable for the work performed therein.
Competing interests. The authors declare no competing interests.
Funding. This research has been conducted within the framework of the research network CeLyA (Lyon Center of Acoustics ANR-10-LABX-60). It has been funded by the LabEx CeLyA (PhD stipend to L.P.), the CNRS, the University of Lyon/Saint-Etienne, and the Institut universitaire de France (N.M.).
Acknowledgements. The authors are grateful to the zoo Planète Crocodiles for providing live animals, to E. Parizet (LVA-INSA) for providing access to the INSA anechoic room, to G. Regnault for help during the acquisitions and to M. Lavandier and T. Leclère for providing part of the Matlab code used in the study.

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
