## [Reviewer comments · Royal Society Open Science]

Review History

RSOS-190423.R0 (Original submission)

Review form: Reviewer 1

Is the manuscript scientifically sound in its present form?

Yes

Are the interpretations and conclusions justified by the results?

Yes

Is the language acceptable?

Yes

Is it clear how to access all supporting data?

Yes

Do you have any ethical concerns with this paper?

No

Have you any concerns about statistical analyses in this paper?

No

Recommendation?

Accept with minor revision (please list in comments)

Comments to the Author(s)

Papet et al. conducted detailed acoustic measurements on a living Nile crocodile and a broad-snouted caiman, as well as on some Nile crocodile skulls to investigate what sound localizations cues could be provided by the head anatomy and the position of the eardrums. They found that the head morphology contributes to the creation of sound location cues, that the same applies to postures in water and on land, and how the interaural canal influences these phenomena.

This paper provides solid, detailed, and scientifically outstanding data. The topic is highly technical, yet the authors manage to include explanations which make the material accessible to readers less familiar with the physics involved. The findings are by their nature descriptive and the authors provide a well-illustrated and highlighted summary.

I only have very few comments and suggestions:

Page 3:

Line 21: I think an "and" is missing after "first),"

Line 36-41: This is confusing me a bit; the English is not very clear. I would suggest the following changes: L36-37: "By combining results from passive acoustic experiments and measurements...", L39: instead of "be lead" rather "focus", L40: in natural positions in water, and on an extended frequency range.

L43-44: "...they mostly come on land to regulate..."

Page 4:

L14-15: Are these the dorsal cranial length (DCL) measurements? Even so, I would argue that a DCL of 22.7 cm is not an adult Nile crocodile, maximally a subadult.

L28-29: If possible please provide the ages, and sizes of the animals. Particularly the head length would be important in the context of the paper.

L39-40: Were the animals actually trained to lay still, e.g. rewarded for staying still? Or were they simply habituated to the procedure?

L43-45: "Effects" could only be noted if the animals could be compared to untested animals. The authors probably mean to say that the animals behaved similarly to the time before testing.

L52: Here and throughout the text, consider to change "Ground condition" to "Land condition". The dichotomy is between land and water; not ground and water. A body of water has a ground.

Page 9:

L47: "such as in" might be misleading here.

Page 10:

L14-15: The sentence starting with “If” confuses me. Please elaborate or rephrase.

Review form: Reviewer 2

Is the manuscript scientifically sound in its present form?

Yes

Are the interpretations and conclusions justified by the results?

Yes

Is the language acceptable?

Yes

Is it clear how to access all supporting data?

Yes

Do you have any ethical concerns with this paper?

No

Have you any concerns about statistical analyses in this paper?

No

Recommendation?

Accept with minor revision (please list in comments)

Comments to the Author(s)

Review of manuscript ‘Influence of head morphology and natural postures on sound localization cues in crocodylians’ (RSOS-190423) by L. Papet et al.

This is an interesting and well-written manuscript that should be published in RS Open Science following several clarifications and additional points. The study is technically elegant, the writing is clear and organized well, and the subject concerns animals that do hear but have received relatively little attention in bioacoustic research. The points below concern ways for making the study more clear, particularly for non-specialists, and additional questions that readers are likely to have.

The authors should emphasize that acquiring information indicative of the location of a sound source (ILD, ITD, monaural cues) does not necessarily mean that such information is actually used by the animal. There is a hint of this awareness in the very end of the discussion, but the authors should be more direct earlier in the paper. An excellent example of this issue was found in the highly developed color vision of mantis shrimp : despite their ability to resolve very small wavelength differences, the ability is not used behaviorally, as in predation and mating. The subject of monaural cues is most interesting, but it needs a few clarifications, notably for non-specialists. These cues, generally associated with the external ear, provide a different spectral profile for sound perceived in a given ear depending on whether the sound source is ipsilateral or contralateral to that ear. The profile differences often entail ‘notches’ at certain frequencies for sound arriving contralaterally, and it can allow an animal that is deaf in one ear to nonetheless localize a sound source. But in intact animals – as the crocodylians in the study were – the cues are really binaural, since the spectral profiles will be available from both ears and can be

compared. Thus, cues from the external ear, which are monaural in a 'one-eared' animal, can serve to enhance ILD and ITD differences in a binaural context.

Can a verbal description be given of the spectral profiles that indicate an ipsilateral or a contralateral sound source? Does this 'monaural' mechanism work better for some sound frequencies than others? Would learning play a role in using monaural cues?

The authors note that a salient feature of crocodylian biology is their amphibious activity. Their study focused on hearing in air, and one wonders whether hearing directionality may also function in water (below the surface)? Do crocodylians need to rely on such hearing for predation? For localizing conspecifics?

The authors note that hearing, and directional hearing in particular, may have influenced the evolution of head morphology in crocodylians. How realistic is this claim given the importance of head morphology (jaw articulations, etc.) for predation? How important is hearing, and specifically directional hearing, for predation in crocodylians? Some natural history information would be helpful.

Decision letter (RSOS-190423.R0)

21-May-2019

Dear Mr Papet

On behalf of the Editors, I am pleased to inform you that your Manuscript RSOS-190423 entitled "Influence of head morphology and natural postures on sound localization cues in crocodylians" has been accepted for publication in Royal Society Open Science subject to minor revision in accordance with the referee suggestions. Please find the referees' comments at the end of this email.

The reviewers and handling editors have recommended publication, but also suggest some minor revisions to your manuscript. Therefore, I invite you to respond to the comments and revise your manuscript.

- Ethics statement

- Data accessibility

<http://datadryad.org/submit?journalID=RSOS&manu=RSOS-190423>

- **Competing interests**

- **Authors' contributions**

- **Acknowledgements**

- **Funding statement**

Because the schedule for publication is very tight, it is a condition of publication that you submit the revised version of your manuscript before 30-May-2019. Please note that the revision deadline will expire at 00.00am on this date. If you do not think you will be able to meet this date please let me know immediately.

on behalf of Dr Claudia Wascher (Associate Editor) and Kevin Padian (Subject Editor)
 openscience@royalsociety.org

Associate Editor Comments to Author (Dr Claudia Wascher):

Associate Editor: 1

Comments to the Author:

The presented paper investigates the influence of head morphology and natural postures on sound localization cues in crocodylians. The results suggest that crocodylian head morphology is adapted to acquire reliable localization cues from sound sources. The reviewers recommend accept with minor revisions.

Reviewer comments to Author:

Reviewer: 1

Comments to the Author(s)

Papet et al. conducted detailed acoustic measurements on a living Nile crocodile and a broad-snouted caiman, as well as on some Nile crocodile skulls to investigate what sound localization cues could be provided by the head anatomy and the position of the eardrums. They found that the head morphology contributes to the creation of sound location cues, that the same applies to postures in water and on land, and how the interaural canal influences these phenomena.

This paper provides solid, detailed, and scientifically outstanding data. The topic is highly technical, yet the authors manage to include explanations which make the material accessible to readers less familiar with the physics involved. The findings are by their nature descriptive and the authors provide a well-illustrated and highlighted summary.

I only have very few comments and suggestions:

Page 3:

Line 21: I think an "and" is missing after "first),"

Line 36-41: This is confusing me a bit; the English is not very clear. I would suggest the following changes: L36-37: "By combining results from passive acoustic experiments and measurements...", L39: instead of "be lead" rather "focus", L40: in natural positions in water, and on an extended frequency range.

L43-44: "...they mostly come on land to regulate..."

Page 4:

L14-15: Are these the dorsal cranial length (DCL) measurements? Even so, I would argue that a DCL of 22.7 cm is not an adult Nile crocodile, maximally a subadult.

L28-29: If possible please provide the ages, and sizes of the animals. Particularly the head length would be important in the context of the paper.

L39-40: Were the animals actually trained to lay still, e.g. rewarded for staying still? Or were they simply habituated to the procedure?

L43-45: "Effects" could only be noted if the animals could be compared to untested animals. The authors probably mean to say that the animals behaved similarly to the time before testing.

L52: Here and throughout the text, consider to change "Ground condition" to "Land condition". The dichotomy is between land and water; not ground and water. A body of water has a ground.

Page 9:

L47: "such as in" might be misleading here.

Page 10:

L14-15: The sentence starting with "If" confuses me. Please elaborate or rephrase.

Reviewer: 2

Comments to the Author(s)

Review of manuscript 'Influence of head morphology and natural postures on sound localization cues in crocodylians' (RSOS-190423) by L. Papet et al.

This is an interesting and well-written manuscript that should be published in RS Open Science following several clarifications and additional points. The study is technically elegant, the writing is clear and organized well, and the subject concerns animals that do hear but have received relatively little attention in bioacoustic research. The points below concern ways for making the study more clear, particularly for non-specialists, and additional questions that readers are likely to have.

The authors should emphasize that acquiring information indicative of the location of a sound source (ILD, ITD, monaural cues) does not necessarily mean that such information is actually used by the animal. There is a hint of this awareness in the very end of the discussion, but the authors should be more direct earlier in the paper. An excellent example of this issue was found in the highly developed color vision of mantis shrimp: despite their ability to resolve very small wavelength differences, the ability is not used behaviorally, as in predation and mating. The subject of monaural cues is most interesting, but it needs a few clarifications, notably for non-specialists. These cues, generally associated with the external ear, provide a different spectral profile for sound perceived in a given ear depending on whether the sound source is ipsilateral or contralateral to that ear. The profile differences often entail 'notches' at certain frequencies for sound arriving contralaterally, and it can allow an animal that is deaf in one ear to nonetheless localize a sound source. But in intact animals – as the crocodylians in the study were – the cues are really binaural, since the spectral profiles will be available from both ears and can be compared. Thus, cues from the external ear, which are monaural in a 'one-eared' animal, can serve to enhance ILD and ITD differences in a binaural context.

Can a verbal description be given of the spectral profiles that indicate an ipsilateral or a contralateral sound source? Does this 'monaural' mechanism work better for some sound frequencies than others? Would learning play a role in using monaural cues?

The authors note that a salient feature of crocodylian biology is their amphibious activity. Their study focused on hearing in air, and one wonders whether hearing directionality may also function in water (below the surface)? Do crocodylians need to rely on such hearing for predation? For localizing conspecifics?

The authors note that hearing, and directional hearing in particular, may have influenced the evolution of head morphology in crocodylians. How realistic is this claim given the importance of head morphology (jaw articulations, etc.) for predation? How important is hearing, and

specifically directional hearing, for predation in crocodilians ? Some natural history information would be helpful.

Author's Response to Decision Letter for (RSOS-190423.R0)

See Appendix A.

Decision letter (RSOS-190423.R1)

31-May-2019

Dear Mr Papet,

I am pleased to inform you that your manuscript entitled "Influence of head morphology and natural postures on sound localization cues in crocodilians" is now accepted for publication in Royal Society Open Science.

on behalf of Dr Claudia Wascher (Associate Editor) and Kevin Padian (Subject Editor)
openscience@royalsociety.org

Subject Areas:

Cognition, Physiology, Biomechanics

Keywords:

Crocodylians, Head-Related Transfer Function (HRTF), Bioacoustics, Sound localization

Author for correspondence:

Léo Papet

e-mail: leo.papet@inserm.fr

Influence of head morphology and natural postures on sound localization cues in crocodylians

L. Papet^{1,2}, N. Grimault¹, N. Boyer², N. Mathevon²

¹Centre de Recherche en Neurosciences de Lyon - Équipe Cognition Auditive et Psychoacoustique, CNRS UMR 5292, Lyon, France.

²Équipe Neuro-Ethologie Sensorielle ENES / NeuroPSI, CNRS UMR 9197 - University of Lyon / Saint-Etienne - Saint-Etienne, France.

As top predators, crocodylians have an acute sense of hearing useful for their social life and for probing their environment in hunting situations. Although previous studies suggest that crocodylians are able to localize the position of a sound source, how they do this remains largely unknown. In this study, we measured the potential monaural sound localization cues (Head-Related Transfer Functions: HRTFs) on alive animals and skulls in two situations, both mimicking natural positions: basking on the **land** and cruising at the interface between air and water. Binaural cues were also estimated by measuring the Interaural Level Differences (ILDs) and the Interaural Time Differences (ITDs). In both conditions, HRTF measurements show large spectral variations (> 10 dB) for high frequencies, depending on the azimuthal angle. These localization cues are influenced by head size and by the internal coupling of the ears. ITDs give reliable information regarding sound-source position for low frequencies, while ILDs are more suitable for frequencies higher than 1.5 kHz. Our results support the hypothesis that crocodylian head morphology is adapted to acquire reliable localization cues from sound sources when outside the water, but also when only a small part of their head is above the air-water interface.

1. Introduction

As top predators, crocodylians have developed fascinating sensory skills: accurate vision in air [1,2], highly developed olfaction [3,4], precise abilities to detect water vibrations [5–7], and an acute sense of hearing [8,9]. Experimental studies, as well as field observations, have demonstrated that the auditory modality is of primary importance for both their social life (e.g. during mating and mother-offspring interactions) and hunting success [10,11]. Although it is known that crocodylians hear sounds over a broad frequency range (from around 300 Hz up to 8 kHz, with a peak in audiogram around 1 kHz [8,9]), and while the functional anatomy of the ear of these archosaurs is similar in many respects to that of birds [12], we have little knowledge on many aspects of their hearing biology. Thus, while some observations have brought evidence for directional hearing in crocodylians, suggesting the existence of specialized adaptations [13–15], how they localize sound sources in their environment remains poorly described.

In birds and mammals including humans, sound-source localization relies on Interaural Time Differences (ITDs: a sound from the left will arrive at the left ear first) and Interaural Level Differences (ILDs: a sound from the left will be louder in the left ear due to the shadowing effect of the head). Moreover, monaural spectral cues are induced by the filtering effect of the head: Head-Related Transfer Functions (HRTFs) result from sounds experiencing spectral modifications during their propagation through and around the head, providing different spectral cues when arriving from different azimuths and elevations [16]. The detection of sounds and the perception of localization cues are often reinforced by some anatomical particularities such as the external ears of mammals [17], and ruff feathers and asymmetrical ears of the barn owl [18]. Remarkably, the hearing apparatus of crocodylians includes a well-developed external ear formed by horny, prominent bone overhanging a muscular ear-lid that protects the eardrum [15,19,20], suggesting a functional role in directional hearing.

Previous studies showed that some directional information is encoded in the auditory nerve of juvenile American alligators *Alligator mississippiensis*. These data support the hypothesis that the acoustic coupling of middle-ear air cavities - an anatomical particularity found in both birds and crocodylians - could enhance localization abilities [21]. By combining results from passive acoustic experiments and measurements of auditory brainstem response to sounds in young alligators, Bierman et al. suggested that HRTFs do not entirely account for the level of directional sensitivity in auditory nerve activity. However this requires further consideration and the current study will focus on larger animals (alive and skulls), in natural positions in water, and on an extended frequency range.

The main characteristic of crocodylian biology is their amphibious way of life. These animals spend most of their active time at the interface between air and water while they mostly come on land to regulate their internal temperature by basking motionless in the sun [22]. In water, the upper part of the head -with the nostrils, eyes and ears- appears above the waterline (figure 1a, b). The acoustic environment of crocodylians may be strongly influenced by this position, as air and water show different sound-propagation properties. As water acts as a reflective surface, it should contribute to the properties of sound waves arriving at the eardrums.

In the present paper, we investigate the acoustic localization cues available to crocodylians, paying specific attention to the effect of the air-water interface, and its interaction with head morphology and head size. We hypothesize that localization cues generated by interactions between sound waves and the head morphology still exist when the animal is at the interface of air and water, in spite of having most of its body concealed underwater. It is worth to note that the presence of such cues does not necessarily induce the behavioral use of the cues by the animals. Using microphones positioned within the ears, we recorded sounds emitted by a source set up at different azimuths from the animal's head axis. We then measured HRTFs, which characterize

the transfer function of the spatial acoustic filter created by the head and body of the animal. This method has commonly been used in humans and other species (e.g. mammals [23,24], alligators [21] and birds [18]). As HRTFs are related to the complex absorption and reflection pattern of acoustical waveforms by the head and body, we compared HRTFs obtained with the crocodile positioned at the interface of air and water with those obtained with the animal's whole body outside water. We further estimated the potential influence of species-specific head shape on HRTFs by performing the experiment on individuals from two different species: the Nile crocodile *Crocodylus niloticus* and the broad-snouted caiman *Caiman latirostris*. To assess the impact of head size, we measured HRTFs on three skulls of different **rostral snout** lengths: 6.9 cm, 16.5 cm and 22.7 cm, corresponding respectively to a juvenile, a young and **subadult** Nile crocodile. Beside HRTF monaural cues, the binaural cues ITDs (Interaural Time Differences) and ILDs (Interaural Level Differences) are also well known to facilitate sound-source localization. We computed ITDs by comparing the waves' arrival time between the two ears, and ILDs by calculating the difference of sound pressure level measured at the right and left ears. **Overall, our results may suggest that the shape of the crocodilian head induces both monaural (HRTF) and binaural (ITD, ILD) sound localization cues even when the head is mostly concealed in water.**

2. Material and methods

(a) Experimental models

We assessed acoustic localization cues on two living animals (see pictures on figure 2) and three skulls (see pictures on figure 3). The animals were one broad-snouted caiman *Caiman latirostris* (**rostral snout length: 4.4 cm, 2 years old**) and one Nile crocodile *Crocodylus niloticus* (**rostral snout length: 6.9 cm, 2 years old**), provided by the zoo Planète Crocodiles (Civaux, France). Both individuals were accommodated in the ENES lab in dedicated areas. These species show strong differences in head morphology (Erickson et al. [25]) that are well-illustrated by the rostral proportion: *C. latirostris* has the broadest snout, whereas the rostral proportion of *C. niloticus* is just above the average of the 23 crocodilian species. In addition, we measured monaural and binaural cues on three skulls of Nile crocodiles (**rostral snout length** from 6.9 cm, 16.5 cm and 22.7 cm).

(b) Animal condition

A critical point with the living individuals was to prevent movements while minimizing stress during HRTF measurements. Three months prior to the experiment, both animals were **habituated** to remain motionless on a board for 30 minutes. The animal position was further secured by straps. This procedure allowed avoiding anaesthesia, which is difficult to master in crocodilians and could have been a survival risk [22]. During the experiments, we (NB) continuously assessed the stress level of the animal by observing its pupillary dilatation and behaviour (escape attempts). During the weeks following the experiments, the condition of both animals was carefully monitored in terms of growth parameters and behaviour (e.g. food intake), and **the animals behaved similarly to the time before testing.**

(c) Signal acquisition for HRTFs measurements

We measured HRTFs in two conditions mimicking biologically relevant situations (figure 1):

1) '**Land** condition': The animal was placed on a board lying on the **land** of a semi-anechoic room (LVA-INSA Lyon: background noise level = 20 ± 1 dB SPL; reverberation time = 0.1 ± 0.1 s), mimicking a position frequently used by crocodilians when basking (figure 1a, c and e). The ground of the semi-anechoic room can be considered as perfectly reflective to acoustic waves.

2) 'Interface condition': The animal was placed in water, with its nostrils, eyes and ears just above the waterline (figure 1b, d, f). This condition mimicked the natural position of an animal in water, e.g. when cruising, ambushing prey or protecting its young. In this position, the water surface was also considered to be fully reverberant, thanks to the short distance between the acoustic source and the microphones.

Under both conditions, the sound source (loudspeaker) was positioned 1 meter from the centre of the head of the animal (defined as the point equidistant between the two ears, see figure 1), with 0° elevation. A rotation of the animal's body along its antero-posterior axis enabled measurements of HRTFs in a 2D plane between -90 and $+90^\circ$ in 5° increments. The sound-emitting equipment was composed of a sound card (Presonus, Audiobox 44-VSL) connected via an amplifier (Yamaha, AX-397) to the loudspeaker (AudioPro loudspeaker, Bravo Allroom Sat). The loudspeaker was hung just above the ground or the waterline to avoid acoustic coupling with the ground or the water. The centre of the medium loudspeaker membrane was placed at the same height as the microphones that were inside the ears.

The experiments were always performed with the ears out of the water. We placed one microphone (Knowles, FG-23329-P07) inside the cavity of each ear, behind the ear-lid and close to, but without touching, the tympanic membrane. This allowed simultaneous recording of the sounds arriving at both right and left ears. The recording equipment was composed of both left and right microphones connected to the input channels of the sound card (sampling frequency = 44.1 Hz).

The emitted signal was a logarithmic sine sweep (frequency range = 20-10000 Hz; duration = 5 s; intensity level = 80 ± 0.5 dB SPL). The frequency range was chosen to mostly cover the hearing range of crocodylians, which is mainly centred on 1-2 kHz [8,9]. For analysis purposes, we used a Matlab code to synchronise in time the source signal with the two recorded microphonic signals.

Prior each measurement, we calibrated the broadcast signals in the absence of any animal, with both microphones placed at the virtual centre of the crocodile's head. This calibration was a necessary step to take into account the properties of the sound-producing and recording equipment ('transfer function' due to material gains, frequency responses, etc.) in HRTFs measurements. Recorded signals were averaged ($n=10$) and used as a reference to compensate for this transfer function.

In addition to HRTF calculations, we measured Interaural Time Differences (ITDs) and Interaural Level Differences (ILDs) using the out-of-water set-up. First, we sent an impulse signal (pulse) to estimate the broadband ITD. We then broadcast 500 ms pure tones at 125, 250, 500 and 1000 Hz. Since Carr et al. [26] demonstrated that in alligators ITDs are not coded in the auditory nerve for frequencies over 1 kHz, we chose to consider only frequencies below 1 kHz. The ITD corresponds to the value of τ maximising the cross correlation between left and right ears $\int_{-\infty}^{+\infty} s_L(t)s_R^*(t-\tau)dt$, where t is the time and τ is the time delay between left and right microphonic signals ($s_L(t)$ and $s_R(t)$, respectively). Assuming symmetry of the head, ITDs were normalized to the 0° value. Directly from HRTF measurements and because of the assumption of symmetry to the normal incidence, the ILD can be calculated for a given frequency and azimuth as:

$$ILD(f, \theta) = H(f, \theta) - H(f, -\theta), \quad (2.1)$$

where f is the frequency, θ the azimuth of the sound source and $H(f, \theta)$ the Head-Related Transfer Function.

(d) Signal processing

To avoid clipping, we applied a Hanning ramp (501 points) at the onset and at the end of recorded microphonic signals, and normalized all recorded signals by the root-mean square amplitude of the normal incidence signal (left and right channels independently).

The spectrum of the recorded microphonic signals (R) within the ear can be expressed as a linear combination of frequency (f), sound source azimuth (θ), elevation (ϕ) and microphone position (\vec{x}), such as:

$$R(f, \theta, \phi, \vec{x}) = S(f) \times H(f, \theta, \phi) \times \mu(f, \vec{x}), \quad (2.2)$$

where $S(f)$ is the calibration signal, $H(f)$ is the Head-Related Transfer Function and $\mu(f, \vec{x})$ is the contribution of the microphone position (adapted from [16]).

In our experiments, elevation was maintained at 0° . HRTFs thus depend only on the sound source azimuth and the sound frequency. One caveat concerns the in-ear position of the microphones: as they were placed under the (opaque) ear-lids, their position could not be perfectly assessed and could be slightly different between left and right ears. We took into account this potential issue by performing two methodological steps. First, the position of the microphone was carefully controlled to be as reliable as possible. Second, we used a normalization method adapted from that developed by Middlebrooks [16] to reduce the effect of microphone position in the human ear canal when measuring HRTF. Briefly speaking, the measured microphonic signal was normalized by the average of all microphonic signals ($\mu(f, \vec{x})$), depending only on the frequency and the position of the microphone. The HRTF was then supposed to depend only on the frequency and the azimuth of the sound source, as follows:

$$H(f, \theta) = \frac{R(f, \theta, \vec{x})}{S(f) \times \mu(f, \vec{x})}, \text{ with: } \mu(f, \vec{x}) = \frac{1}{S(f)} \frac{1}{N} \sum_{i=1}^N R_i(f, \theta_i, \vec{x}), \quad (2.3)$$

where N is the number of microphonic signals.

To limit the error in HRTF estimations, we considered in this study each head as symmetrical and thus averaged the HRTFs simultaneously measured in the right and left ears. To limit discontinuities in HRTF measurements along angular and frequential axes, we applied a smoothing procedure based on a two-dimensional floating Gaussian window normalised in amplitude with a 5 points width in azimuth and a logarithmically varying width in frequency (3 Hz at $f = 20$ Hz and 1 kHz at $f = 10$ kHz).

Finally, we calculated a Potential Localization Level (PLL) based on HRTF and ILD as follows:

$$PLL_H(f) = 20 \times \log_{10} \left(\frac{1}{N_\theta} \sum_{\theta=-90}^{90} \left| \frac{\partial H(f, \theta)}{\partial \theta} \right| \right), \quad (2.4)$$

and,

$$PLL_{ILD}(f) = 20 \times \log_{10} \left(\frac{1}{N_\theta} \sum_{\theta=-90}^{90} \left| \frac{\partial ILD(f, \theta)}{\partial \theta} \right| \right), \quad (2.5)$$

with N_θ is the number of azimuth positions (in here θ varies between -90 and 90° with a step of 5° , so: $N_\theta = 37$). $PLL_H(f)$ and $PLL_{ILD}(f)$ are expressed in *dB*. The PLLs are computed to facilitate the comparison of HRTFs and ILDs between the different conditions. Classically, sound localization cues are considered efficient when varying monotonously according to the azimuth of the sound source. For each frequency, the PLL corresponds to the average of the variation of $H(f, \theta)$ or $ILD(f, \theta)$ according to θ in *dB*. So, the Potential Localization Level is a cumulated measurement of variations of monaural cues across azimuth and a high PLL indicates a strong variation of the considered cue with the position of the source.

3. Results

(a) HRTF cues in **land-** and air-water interface conditions

The HRTFs measured in the awake animals in both **land-** and interface conditions are displayed in figure 2 (panels a-d). The HRTF intensity level is coded by an arbitrary coloured dB scale (from -12 to 8 dB arb., with contour lines representing 5 dB intervals), and expressed as both a function of frequency (20 - 10000 Hz) and of the sound-source azimuth (-90 - $+90$ °). Positive (resp. negative) values of HRTF intensity level induce an amplification (resp. attenuation) of the acoustic field due to the presence of the head of the animal compared to the calibration situation (i.e. with no animal). Positive azimuth angles correspond to sounds recorded from the ipsilateral side, i.e. from the side of the sound source, whereas negative azimuths correspond to the contralateral side, i.e. sounds recorded in the 'acoustic shadow' of the head.

In both **land-** and air-water interface conditions, HRTFs showed similar complex patterns of sound pressure level variations, with high dynamics (20 dB) depending both on sound-source incidence and sound frequency (figure 2a-d). This intensity pattern varied depending on the side: for azimuths smaller than 0° (contralesional side), the sound level measured near the eardrum is negative whereas it appears mainly positive (i.e. amplified) for incidences larger than 0° (ipsilateral side). This main result indicates that the angular position of the sound source influences the spectral cues perceived at the ears' level, suggesting that HRTFs could support sound source localization in both the **land-** and air-water interface conditions. The difference in sound-pressure levels between the ears due to the position of the sound source was, however, mainly present for frequencies above 1 kHz. Below 400 Hz, the sound-pressure level was mostly invariant to sound-source position.

Beside this general picture, HRTFs were characterized by 'bumps' and 'notches' which may increase the locatability of the sound source. Globally, the complexity of the HRTF patterns increases with frequencies (figure 2). As the emitted sound signal showed equal levels across the whole frequency spectrum, the variations of this level are due to the filtering effect of the head. These complex shapes may underlie the complexity of the acoustic field surrounding the animals' head. For instance, when the Nile crocodile was in the **land** condition (figure 2a), the sound-pressure level near 6 kHz was strongly influenced by the source angle (variations from -11 dB up to 6 dB), with a marked area of low levels when the source was positioned at -40° .

Overall, these results suggest that HRTF localization cues are already present near 1 kHz in both **land** and interface conditions, and become more important when sound frequency increases. This is further illustrated by the Potential Localization Levels (PLLs) displayed on figure 2e and f. PLLs represent the amount of external acoustic localization cues measured at the tympanic membrane level (see Methods). In accordance with HRTF results, PLLs increase exponentially with sound frequency (i.e. linearly with the logarithm of the frequency). They look very similar in both species and conditions (**land** condition: 2.7 dB/octave for the crocodile and 2.9 dB/octave for the caiman; air-water interface condition: 2.4 and 2.5 dB/octave, respectively).

(b) Impact of skull size and acoustic coupling through the interaural canal

The HRTFs measured on the three skulls of *Crocodylus niloticus* are displayed on figure 3a-c. Compared to those obtained in alive animals, HRTFs show mainly the same pattern with more complex cues in high frequencies and a higher intensity level in the ipsilateral side. In the low frequency range ($f < 1$ kHz), it can be noticed that HRTF patterns appear more homogeneous in contrast to those measured on alive animals. Nevertheless, the main difference is the presence of a 'crescent shaped' area of low sound-pressure level (underlined by the white continuous line on the HRTF color maps in figure 3a-c). This pattern, consistently found in the three skulls, is

included in a frequency band which is directly dependent on the skull size (3.5 - 6.5 kHz, 2.5 - 6.5 kHz and 1.7 - 5.5 kHz for the small, medium and large skulls, respectively), and may be due to destructive interferences caused by the interaural canal. Considering one ear, the difference of pathway between the direct wave and the wave arriving from the other ear through the interaural canal induces a phase difference Φ :

$$\Phi = 2\pi f \frac{\delta}{c}, \quad (3.1)$$

where δ is the difference of pathway in meters, c the sound velocity and f the frequency. From 3.1, if $\Phi = \pi [2\pi]$, we can compute the frequencies corresponding to destructive interferences inside the interaural canal f_{ic} :

$$f_{ic} = \frac{c}{2\delta}. \quad (3.2)$$

This simple geometrical model plotted on figure 3a-c indeed suggests that this pattern is due to the interaural canal.

In line with what we found in alive animals, the effect of sound-source position on HRTFs varied as a function of skull size. For instance, a 2 kHz sound produces complex level variation that depended on the sound-source position in the larger skull (between -4 and +4 dB), while the sound-pressure level remained constant for all azimuths (0 dB) for the small and medium skulls. Moreover, the maximum sound-pressure level areas measured on the ipsilateral side extended to a lower frequency range when skull size increased. For an incidence of 90° , the 3 dB contour line starts at 4040 Hz, 3050 Hz and 2120 Hz for the small, medium and large skulls, respectively.

In skulls measurements, the Potential Localization Levels (PLLs) did not increase linearly with the logarithm of the frequency (figure 3d) and it is not relevant to model its evolution using linear regression. In the low frequency range ($f < 1\text{kHz}$), the PLLs remained almost steady around -60 dB. For frequencies larger than 1 kHz, the PLLs increased with frequency in line with the complexity of HRTF patterns. In skulls, HRTFs depended on the global shape of the head but were also modified by the interaural canal, causing a non linear evolution of the PLLs.

(c) Binaural cues

Based on formula 2.1, PLLs were computed from ILDs and calculated using 2.5 (figure 4a and b). As displayed in figure 4a-c, the PLLs calculated for alive animals increased monotonically with the logarithm of frequency, with no noticeable impact of species or condition. The effect of head size is emphasised by PLLs calculated from skulls. Thus for a 1 kHz sound, PLL is equal to -52, -49 and -38 dB for the small, medium and large skulls, respectively (figure 4c). ILDs are stronger for frequencies higher than 1 kHz, with a sudden increase of the slope near 1 kHz of PLLs measured on skulls.

We assessed ITDs on the medium size skull in only two conditions: the interaural canal was either obstructed with an adhesive or opened (figure 4d). The ITDs results are very close to those obtained by Carr et al. on *Alligator mississippiensis* [26]: ITDs are symmetrical to the normal incidence and vary monotonically with the position of the sound source. When open (figure 4 right side of panel d), the interaural canal led to a decrease in ITDs, offering a shorter pathway to acoustic waves. As a result, the maximum of ITD (at 90°) is decreased by about $50 \mu\text{s}$ when the canal is let free, independently from the frequency.

4. Discussion

Our study presents evidence that the morphology of the head of crocodilians induces monaural and binaural acoustic cues available to the animal and potentially useful to obtain information on the position of a sound source. These cues are still present when most of the animal body is underwater, suggesting that the well-developed external ear formed by the horny and prominent bone is efficient to provide external localization cues both on the land and at the interface. This could represent an evolutionary adaptation to the peculiar amphibious behaviour of crocodilians.

Spectral monaural cues (HRTF) are present mainly for frequencies higher than 1 kHz. Their saliency increases with sound frequency, and they are strongly influenced by head size, with a shift to a lower frequency range in larger heads. Interestingly, we found that HRTFs cues are very similar in both 'land-' and 'air-water interface' conditions. This suggests that the ability to use monaural cues for sound source localization may be alike in both conditions, despite only part of the crocodile's head being exposed in the interface condition. Our investigations on skulls underline the importance of the interaural canal, which creates destructive interferences that may further facilitate sound-source localization. In addition to highlighting HRTF cues, we confirm the presence of ILD cues (mainly for frequencies above 1 kHz) as well as ITD cues (mainly in the low frequency range, below 1 kHz). Altogether, monaural and binaural cues may allow crocodilians to accurately localize the position of a broadband sound source in their environment.

Overall, our results are consistent with Bierman et al. [21]. These authors concluded from their measurements of external cues that the "*acoustic space cues generated by the external morphology of the animal are not sufficient to generate location cues that match physiological sensitivity*" [21]. Indeed, they demonstrated that the level of physiological sensitivity is due to the contributions of the sound localization cues, the internal coupling of middle ears and the directionality of the eardrum. However, Bierman et al. worked on very young alligators, with an interaural distance of 2.25 ± 0.2 cm, whereas our animals showed interaural distances of 3.9 cm and 4.7 cm, respectively, for the broad-snouted caiman and the Nile crocodile. The juvenile skull of *Crocodylus niloticus* had interaural distances comparable to those of the alligators studied by Bierman et al. (2.4 cm). Besides, they conducted their experiments for frequencies lower than 4 kHz and they obtained a maximum variation of 8 dB (for a frequency of 2 kHz). In the same frequency range, we measured a slightly higher dynamics of 9 dB maximum (cf. figures 2 and 3). In the present study, the frequency span is increased up to 10 kHz because for high intensity sounds, the high frequencies may be of interest for crocodilians. Indeed we cannot exclude the potential use of frequencies higher than 2 kHz, and as we have strong dynamics (up to 20 dB), this frequency region may be relevant for sound localization. Assuming the small size of the animals and that the external cues are shifted to low frequency when size increases (figure 3), our results extend Bierman's work and is coherent with their results.

Given that the crocodilians' audiogram is centred on the lower part of the frequency spectrum (1-2 kHz, [8,9,27]) and that the external localization cues increase with frequency, the biological relevance of the HRTFs has to be discussed. The frequency span used in our study was chosen to widely cover the audition curve of crocodilians, and to illustrate the low-frequency shift phenomena induced by size. If we consider the maximum of sensitivity of audition in crocodilians (i.e. 1.5 kHz), the dynamics measured on external cues is more than 5 dB (figure 2), which is potentially sufficient for sound localization. In blackbirds, pigeons and in sea lions the minimum detectable binaural level difference is around 3dB [28,29] whereas it is 1 dB in humans [30].

Our study focused on two awake juvenile animals belonging to two different species. However, the results could be extended to larger animals. Although we found a comparable amount of potential localization cues in both experimental animals, slight differences in HRTFs between species were visible in the upper part of the frequency spectrum. Studies in humans have

shown that individual morphology (i.e. rather small morphological differences) may significantly influence HRTFs [31]. As both our animals had comparable body sizes, these differences may have been induced by their respective head morphology (e.g. the Nile crocodile has a much more slender snout than the caiman). In larger animals, these slight differences should be enhanced and shifted towards lower frequencies. As a consequence, they may significantly influence HRTFs within their hearing range.

Besides HRTFs, the internal acoustic coupling may increase sound localization because two waves arrive at the same eardrum: one from the outside and one from the inside through the interaural canal. The influences of this coupling were shown using a method of geometrical acoustics on skulls (figure 3). The interaural canal also influences the ITD by decreasing its value from 50 μ s. **While this result looks opposite to previous published data [32,33], it is explained by the fact that we consider the interaural canal on skulls but without any soft tissue (such as eardrums).** The canal acts as a wave-guide without any obstacle to the propagation of the acoustic wave. Bierman et al. demonstrated the implication of the directionality of the eardrums on physiological relevance of ITD computation thanks to a laser vibrometry method [21].

ITDs are large in the low-frequency region (from 100 to 1000 Hz), whereas ILDs predominate in the high frequencies region ($f > 1500$ Hz). In his duplex theory developed for humans [34], Lord Rayleigh asserted that ITDs and ILDs ensure strong localization abilities across the full auditory spectrum thanks to an overlap of both binaural cues between 1 and 1.5 kHz [35]. This theory cannot be applied to all vertebrates: as an example, the barn owl combines ITD and ILD information in the same frequency range (between 3 and 10 kHz) to localise respectively the azimuth and the elevation [36,37]. In another way, crocodylians might also qualify this theory. Carr et al. [26] demonstrated that ITD is not neurally processed for frequencies higher than 1 kHz because of a lack of phase locking. As a consequence, ITDs and ILDs are weak in the band between 1 and 1.5 kHz, creating a lack of localization cues in this frequency region, at least for juvenile and medium-sized crocodylians. The maximum of vibration of the eardrum was measured in this precise bandwidth [21], which could be used to compensate this 'gap' of localization.

As a conclusion, our results establish a strong background regarding the acoustic cues available to crocodylians when they localize a sound source in their environment -a frequent situation in several behavioural contexts, from predation to caring for the young. **Our study focused on aerial hearing. However, crocodylians are amphibious animals and previous studies suggested a fairly good underwater hearing sensitivity. It would thus be interesting to investigate sound localization cues in an underwater context. Finally, sound localization abilities remain also poorly investigated and, even if a few behavioural observations in crocodylians were published [13,38], the behavioural relevance of sound localization cues needs to be tested in following research.**

Ethics

All the experiments were conducted in the ENES lab agreement number (D 42-218-0901). The animals were under the responsibility of Nicolas Boyer, who is officially certificated to work with crocodylians (certificate of capacity).

Data accessibility

The datasets and program files supporting this article have been uploaded on the Zenodo platform (zenodo.org/record/2572624#.XGv8UbhCdQI - DOI: 10.5281/zenodo.2572624, [39]) with an embargoed access. Moreover, the same dataset can be downloaded by the editors and reviewers using the link <https://dl.univ-lyon1.fr/bv69hsj3> with the password: RawDataPapet.

Authors' contributions

L.P. designed the study, conducted the experiments and ran the analysis. N.G. and N.M. participated in data collection and analysis. N.B. participated in data collection and was in charge of the manipulation and of the well-being of the animals before, during and after the experiments. L.P., N.G. and N.M. wrote the manuscript. All authors gave final approval for publication and agree to be held accountable for the work performed therein.

Competing interests

The authors declare no competing interests.

Funding

This research has been conducted within the framework of the research network CeLyA (Lyon Center of Acoustics ANR-10-LABX-60). It has been funded by the LabEx CeLyA (PhD stipend to LP), the CNRS, the University of Lyon/Saint-Etienne, and the Institut universitaire de France (NM).

Acknowledgements

The authors are grateful to the zoo Planète crocodiles for providing alive animals, to E. Parizet (LVA-INSA) for providing access to the INSA anechoic room, to G. Regnault for helping during the acquisitions and to M. Lavandier and T. Leclère for providing part of the Matlab code used in the study.

Figure 1. Experimental set-up used to measure HRTFs localization cues. Two natural postures of crocodilians considered in the present study: (a) on the **land** and (b) at the interface between air and water. Cross-section and top view of the set-up in the **land** condition (c) and (e), and at the interface (d) and (f).

Figure 2. Head-Related Transfer Functions measured on awake animals in two natural positions. (a) HRTF measured on *Crocodylus niloticus* in the **land** condition. (b) HRTF measured on *Caiman latirostris* in the **land** condition. (c) HRTF measured on *Crocodylus niloticus* in the **interface** condition. (d) HRTF measured on *Caiman latirostris* in the **interface** condition. The considered animal and configuration are represented above each panel at scale 1 / 12. (e) Potential Localization Level measured on *Crocodylus niloticus* in **land** (blue) and **interface** (red) conditions. (f) Potential Localization Level measured on *Caiman latirostris* in **land** (blue) and **interface** (red) conditions. (e), (f) Solid lines correspond to raw data and dashed lines are linear regressions.

Figure 3. Head-Related Transfer Functions measured on three skulls of different sizes. (a), (b) and (c): HRTFs measured on three *Crocodylus niloticus* skulls of different lengths: 6.9 cm, 16.5 cm, and 22.7 cm, respectively. The solid white line represents the destructive interferences based on a simple geometrical model of the path difference. The considered skull is represented above each panel at scale 1 / 12. (d) Potential Localization Level computed for the three skulls: 6.9 cm (blue), 16.5 cm (purple), and 22.7 cm (red).

Figure 4. Binaural cues measured on awake animals and skulls of different sizes. Potential Localization Level computed on ILD for *Crocodylus niloticus* (a) and *Caiman latirostris* (b) measured in the **land** (blue) and **interface** (red) situations. Dashed lines in (a) and (b) corresponds to linear regressions. (c) Potential Localization Level computed for ILD for three different sized Nile crocodile skulls: 6.9 cm (blue), 16.5 cm (purple), and 22.7 cm (red). (d) ITD measured for 4 pure tones (125, 250, 500 and 1000 Hz) and for a pulse with the interaural canal blocked (left) and open (right).

References

1. Sillman AJ, Ronan SJ, Loew ER. Histology and microspectrophotometry of the photoreceptors of a crocodilian, *Alligator mississippiensis*. *Proceedings of the Royal Society of London. Series B: Biological Sciences*, 243(1306):93–98, January 1991. (doi:10.1098/rspb.1991.0016)
2. Nagloo N, Collin SP, Hemmi JM, and Nathan S. Hart. Spatial resolving power and spectral sensitivity of the saltwater crocodile, *Crocodylus porosus*, and the freshwater crocodile, *Crocodylus johnstoni*. *The Journal of Experimental Biology*, 219(9):1394–1404, May 2016. (doi:10.1242/jeb.135673)
3. Scott, TP, Weldon PJ. Chemoreception in the feeding behaviour of adult American alligators, *Alligator mississippiensis*. *Animal Behaviour*, 39(2):398–400, February 1990. (doi:10.1016/S0003-3472(05)80887-5)
4. Weldon PJ, Ferguson MWJ. Chemoreception in Crocodilians: Anatomy, Natural History, and Empirical Results. *Brain Behavior Evolution*, 41(3-5):239–245, February 1993. (doi:10.1159/000113845)
5. Fleishman LJ, Stanley Rand A. *Caiman crocodilus* does not require vision for underwater prey capture. *Journal of Herpetology*, 23(3):296, September 1989. (doi:10.2307/1564453)
6. Leitch DB, Catania KC. Structure, innervation and response properties of integumentary sensory organs in crocodilians. *Journal of Experimental Biology*, 215(23):4217–4230, December 2012. (doi:10.1242/jeb.076836)
7. Grap NJ, Monzel AS, Kohl T, Bleckmann H. *Crocodylus niloticus* (Crocodilia) is highly sensitive to water surface waves. *Zoology*, 118(5):320–324, October 2015. (doi:10.1016/j.zool.2015.03.004)
8. Wever EG. Hearing in the Crocodilia. *Proceedings of the National Academy of Sciences*, 68(7):1498–1500, May 1971. (doi:10.1073/pnas.68.7.1498)
9. Higgs D, Brittan-Powell E, Soares D, Souza M, Carr CE, Dooling R, Popper A. Amphibious auditory responses of the American alligator (*Alligator mississippiensis*). *Journal of Comparative Physiology A: Sensory, Neural, and Behavioral Physiology*, 188(3):217–223, April 2002. (doi:10.1007/s00359-002-0296-8)
10. Vergne AL, Avril A, Martin S, Mathevon N. Parent-offspring communication in the Nile crocodile *Crocodylus niloticus*: do newborns' calls show an individual signature? *Naturwissenschaften*, 94(1):49–54, December 2006. (doi:10.1007/s00114-006-0156-4)
11. Vergne AL, Aubin T, Martin S, Mathevon N. Acoustic communication in crocodilians: information encoding and species specificity of juvenile calls. *Animal Cognition*, 15(6):1095–1109, November 2012. (doi:10.1007/s10071-012-0533-7)
12. Vergne AL, Pritz MB, Mathevon N. Acoustic communication in crocodilians: from behaviour to brain. *Biological Reviews*, 84(3):391–411, August 2009. (doi:10.1111/j.1469-185X.2009.00079.x)
13. Beach FA. Responses of Captive Alligators to Auditory Stimulation. *The American Naturalist*, 779(78):481–505, November 1944.
14. Dinets V. Do individual crocodilians adjust their signaling to habitat structure? *Ethology Ecology & Evolution*, 25(2):174–184, April 2013. (doi:10.1080/03949370.2012.744358)
15. Bierman HS, Carr CE. Sound localization in the alligator. *Hearing Research*, 329:11–20, November 2015. (doi:10.1016/j.heares.2015.05.009)
16. Middlebrooks JC, Green DM. Directional dependence of interaural envelope delays. *The Journal of the Acoustical Society of America*, 87(5):2149–2162, May 1990. (doi:10.1121/1.399183)
17. De Mey F, Reijnen J, Peremans H, Otani M, Firzlaff U. Simulated head related transfer function of the phyllostomid bat *Phyllostomus discolor*. *The Journal of the Acoustical Society of America*, 124(4):2123–2132, October 2008. (doi:10.1121/1.2968703)
18. Keller CH, Hartung K, Takahashi TT. Head-related transfer functions of the barn owl: measurement and neural responses. *Hearing Research*, 118(1-2):13–34, April 1998. (doi:10.1016/S0378-5955(98)00014-8)
19. Montefeltro FC, Andrade DV, Larsson HCE. The evolution of the meatal chamber in crocodyliforms. *Journal of Anatomy*, 228(5):838–863, May 2016. (doi:10.1111/joa.12439)
20. Manley, GA Comparative Auditory Neuroscience: Understanding the Evolution and Function of Ears. *Journal of the Association for Research in Otolaryngology*, 18(1):1–24, August 2016. (doi:10.1007/s10162-016-0579-3)
21. Bierman HS, Thornton JL, Jones HG, Koka K, Young BA, Brandt C, Christensen-Dalsgaard J, Carr CE, Tollin DJ. Biophysics of directional hearing in the American alligator (*Alligator mississippiensis*). *Journal of Experimental Biology*, 217(7):1094–1107, April 2014. (doi:10.1242/jeb.092866)

22. Grigg GC, Kirshner D. *Biology and evolution of crocodylians*. Comstock Publishing Associates, a division of Cornell University Press, Ithaca, January 2015.
23. Rébillat M, Benichoux V, Otani M, Keriven R, Brette R. Estimation of the low-frequency components of the head-related transfer functions of animals from photographs. *The Journal of the Acoustical Society of America*, 135(5):2534–2544, May 2014. (doi:10.1121/1.4869087)
24. Koka K, Jones HG, Thornton JL, Lupo JE, Tollin DJ. Sound pressure transformations by the head and pinnae of the adult Chinchilla (*Chinchilla lanigera*). *Hearing Research*, 272(1-2):135–147, February 2011. (doi:10.1016/j.heares.2010.10.007)
25. Erickson GM, Gignac PM, Steppan SJ, Lappin AK, Vliet KA, Brueggen JD, Inouye BD, Kledzik D, Webb GJW. Insights into the Ecology and Evolutionary Success of Crocodylians Revealed through Bite-Force and Tooth-Pressure Experimentation. *PLOS ONE*, 7(3):e31781, March 2012. (doi:10.1371/journal.pone.0031781)
26. Carr CE, Soares D, Smolders J, Simon JZ. Detection of Interaural Time Differences in the Alligator. *Journal of Neuroscience*, 29(25):7978–7990, June 2009. (doi:10.1523/JNEUROSCI.6154-08.2009)
27. Manley GA. Frequency sensitivity of auditory neurons in the Caiman cochlear nucleus. *Zeitschrift für Vergleichende Physiologie* 66(3):251–256, October 1970. (doi:10.1007/BF00297828)
28. Hienz RD, Sinnott JM, Sachs MB. Auditory intensity discrimination in blackbirds and pigeons. *Journal of Comparative and Physiological Psychology*, 94(6):993–1002, February 1980. (doi:10.1037/h0077734)
29. Moore PWB, Schusterman RJ. Discrimination of pure-tone intensities by the California sea lion. *The Journal of the Acoustical Society of America*, 60(6):1405–1407, January 1977. (doi:10.1121/1.2003587)
30. Mills AW. Lateralization of High-Frequency Tones. *The Journal of the Acoustical Society of America*, 32(1):132–134, January 1960. (doi:10.1121/1.1907864)
31. Moller H, Sorensen MF, Jensen CB, Hammershoi D. Binaural techniques: Do we need individual recordings? *Journal of the Audio Engineering Society*, 44(6):451–469, June 1996.
32. Carr CE, Christensen-Dalsgaard J, Bierman H. Coupled ears in lizards and crocodylians. *Biological Cybernetics*, 110(4-5):291–302, October 2016. (doi:10.1007/s00422-016-0698-2)
33. Christensen-Dalsgaard J, Tang Y, Carr CE. Binaural processing by the gecko auditory periphery. *Journal of Neurophysiology*, 105(5):1992–2004, May 2011. (doi:10.1152/jn.00004.2011)
34. Rayleigh L. On the Perception of the Direction of Sound. *Proceedings of the Royal Society A: Mathematical, Physical and Engineering Sciences*, 83(559):61–64, November 1909. (doi:10.1098/rspa.1909.0073)
35. Feddersen WE, Sandel TT, Teas DC, Jeffress LA. Localization of high-frequency tones. *The Journal of the Acoustical Society of America*, 29(9):988–991, September 1957. (doi:10.1121/1.1909356)
36. Coles RB, Guppy A. Directional hearing in the barn owl (*Tyto alba*). *Journal of Comparative Physiology A*, 163(1):117–133, January 1988. (doi:10.1007/BF00612002)
37. Knudsen EL, Konishi M. Mechanisms of sound localization in the barn owl (*Tyto alba*). *Journal of Comparative Physiology A*, 133(1):13–21, March 1979. (doi:10.1007/BF00663106)
- Acoustic Signaling via "Pops". *Proceedings, 13th Western Pacific Acoustics Conference*, 1–6, November 2019.
38. Chabert T, Colin A, Aubin T, Shacks V, Bourquin SL, Elsey RM, Acosta JG, Mathevon N. Size does matter: crocodile mothers react more to the voice of smaller offspring. *Scientific Reports*, 5(1):2045–2322, December 2015. (doi:10.1038/srep15547)
39. Papet L, Grimault N, Boyer N, Mathevon N. Data from: Influence of head morphology and natural postures on sound localization cues in crocodylians, 2019. (doi:10.5281/zenodo.2572624)